# Leveraging Motor Imagery Rehabilitation for Individuals with Disabilities: A Review

**DOI:** 10.3390/healthcare11192653

**Published:** 2023-09-29

**Authors:** Maram Fahaad Almufareh, Sumaira Kausar, Mamoona Humayun, Samabia Tehsin

**Affiliations:** 1Department of Information Systems, College of Computer and Information Sciences, Jouf University, Sakakah 72388, Saudi Arabia; 2Center of Excellence in Artificial Intelligence COE-AI, Department of CS, Bahria University, Islamabad 44000, Pakistan; sumairakausar.buic@bahria.edu.pk (S.K.); stehseen.buic@bahria.edu.pk (S.T.)

**Keywords:** disability, motor imagery, rehabilitation, motor skills

## Abstract

Motor imagery, an intricate cognitive procedure encompassing the mental simulation of motor actions, has surfaced as a potent strategy within the neuro-rehabilitation domain. It presents a non-invasive, economically viable method for facilitating individuals with disabilities in enhancing their motor functionality and regaining self-sufficiency. This manuscript delivers an exhaustive analysis of the significance of motor imagery in augmenting functional rehabilitation for individuals afflicted with physical impairments. It investigates the fundamental mechanisms governing motor imagery, its applications across diverse disability conditions, and the prospective advantages it renders. Moreover, this document addresses the prevailing obstacles and prospective trajectories in this sector, accentuating the necessity for continued investigation and the invention of cutting-edge technologies that optimize the potentiality of motor imagery in aiding disabled persons.

## 1. Introduction

Motor impairments can substantially affect an individual’s quality of life by restricting their capacity to conduct daily tasks, participate in social interactions, and sustain independence. Conventional rehabilitation methods such as physiotherapy and assistive apparatuses have demonstrated efficacy, but often encounter limitations in achieving optimal results [1,2]. In recent years, people have been paying more attention to the idea that imagining themselves doing physical movements could be a helpful addition to rehabilitation for individuals with disabilities [3,4].

By understanding how motor imagery can be valuable in disability rehabilitation, a diverse group of healthcare professionals, researchers, and people with disabilities can work together to use this innovative approach to improve functional outcomes and enhance the overall well-being of individuals with disabilities.

Motor imagery Figure 1, also referred to as mental imagery or kinesthetic imagery, denotes the cognitive procedure of mentally simulating or rehearsing a movement without physically enacting it [4]. It encompasses the formation of vivid and elaborate mental depictions of motor actions, integrating sensory, perceptual, and proprioceptive facets [5]. By engaging in motor imagery, individuals can mentally “practice” movements, triggering neural networks analogous to those involved in actual physical execution [3]. Motor imagery functions through an intricate interplay of neural mechanisms. Neuroimaging studies have disclosed that during motor imagery, corresponding brain regions accountable for planning and implementing physical movements are activated, including the primary motor cortex, premotor regions, supplementary motor area, and parietal areas [6,7]. This activation culminates in the production of internal motor representations which can impact the subsequent performance of motor tasks [8].

Motor imagery is intimately connected to motor learning—the process of attaining and refining motor skills via practice [9]. Empirical evidence indicates that motor imagery can bolster motor learning by facilitating the consolidation and refinement of motor representations [5]. Studies show that through mental rehearsal of movements, individuals can enhance their motor planning, coordination, and precision, resulting in more proficient motor execution [10]. This attribute of motor imagery renders it an invaluable resource in rehabilitation, where motor learning is a crucial component in functional recuperation [11]. This manuscript aims to investigate the utilization of motor imagery in relation to various disability circumstances such as stroke rehabilitation, spinal cord injuries, traumatic brain injuries, Parkinson’s disease, cerebral palsy, and musculoskeletal disorders [6,9]. The advantages of motor imagery in enhancing motor functionality, cognitive capability augmentation, psychological well-being promotion, and fostering neuroplasticity for motor recuperation will be examined. Additionally, the manuscript will delve into diverse motor imagery techniques and training protocols including mental practice, virtual reality applications, biofeedback modalities, and brain–computer interfaces while analyzing their respective strengths and drawbacks. The obstacles and restrictions linked to the integration of motor imagery into clinical practices—variability in imaginative faculty and technological restraints—are also assessed [12]. Conclusively, the manuscript will expound upon future research trajectories encompassing the formulation of tailored training regimens, refinement of neurofeedback methodologies, breakthroughs in both virtual reality and brain–computer interface technologies, and the significance of longitudinal follow-up and persistence within motor imagery interventions [11]. This paper aims to offer a comprehensive understanding of the function of motor imagery in ameliorating motor performance and augmenting the rehabilitation process for individuals with disabilities (having impaired motor functions) [3,4]. By investigating the fundamental mechanisms, scrutinizing its applications across various disability conditions [6,9], and deliberating on prospective advantages and challenges [12], this paper endeavors to illuminate the importance of motor imagery as an inventive strategy for disability rehabilitation [10,11]. Section 2 presents the mechanism of motor imagery, then the physiological mechanism of motor imagery is discussed, which is followed by motor learning and rehabilitation using motor imagery. Then, different protocols are discussed before identifying challenges and opportunities and offering concluding sections.

## 2. Mechanism of Motor Imagery

Motor imagery is a complex cognitive process encompassing the mental simulation and rehearsal of movements without engaging in their physical execution. This allows individuals to generate and manipulate mental representations of motor actions, incorporating sensory, perceptual, and proprioceptive elements. The concept is based on the premise that the brain can produce internal motor representations that share similarities with those activated during actual movement [6,7].

There are several key components involved in motor imagery:

### 2.1. Mental Simulation

This component requires individuals to mentally simulate the performance of a specific movement or action, generating a mental image or representation of themselves executing the desired action. This process necessitates visualization of the body and its respective movements [5].

### 2.2. Kinesthetic Imagery

Kinesthetic imagery pertains to the subjective experience entailing emulation of bodily movement sensations without any physical execution. This component encompasses a sense of body position, movement, and effort, empowering individuals to mentally perceive the kinesthetic sensations concomitant with the intended action [4].

### 2.3. Visual Imagery

This aspect involves the generation of mental images or visual representations pertinent to movement. It encompasses visualization of bodily movements, environmental context, and any relevant objects or obstacles within said environment [9].

### 2.4. Temporal Imagery

Temporal imagery pertains to the process of mentally representing timing and sequencing aspects of movements. Individuals imagine attributes such as duration, rhythm, and coordination of actions, therefore allowing them to mentally practice and refine their motor skills [13].

## 3. Neurophysiological Mechanisms

Exploring the neural mechanisms underpinning motor imagery offers valuable information regarding its influence on motor learning and rehabilitation processes. Neuroimaging studies employing techniques like functional magnetic resonance imaging (fMRI) and electroencephalography (EEG) have elucidated brain regions and networks associated with motor imagery [12,14,15].

Regions such as the primary motor cortex (M1) and premotor cortex, accountable for motor planning and execution, are activated during motor imagery sequences [6,7,14]. These regions generate neural signals analogous to those produced during actual movements. The supplementary motor area (SMA) also participates in this process, contributing to the initiation and coordination of motor plans Figure 2.

Parietal regions like the superior parietal lobule and inferior parietal lobule play a crucial role in sensorimotor integration during motor imagery. These regions combine visual, proprioceptive, and kinesthetic information to formulate a coherent mental representation of the movement [10]. The Mirror Neuron System (MNS) constitutes a fundamental component in the realm of motor imagery. Initially identified in macaque monkeys and subsequently in humans, mirror neurons represent a unique class of neurons that exhibit activation both during the execution of a specific action by an individual and while observing another individual performing the same action. The MNS contributes significantly to motor imitation and comprehending the intentions of others. Moreover, it is postulated to play an essential role in motor imagery by facilitating the internal simulation of observed movements [8].

**Figure 2 healthcare-11-02653-f002:**
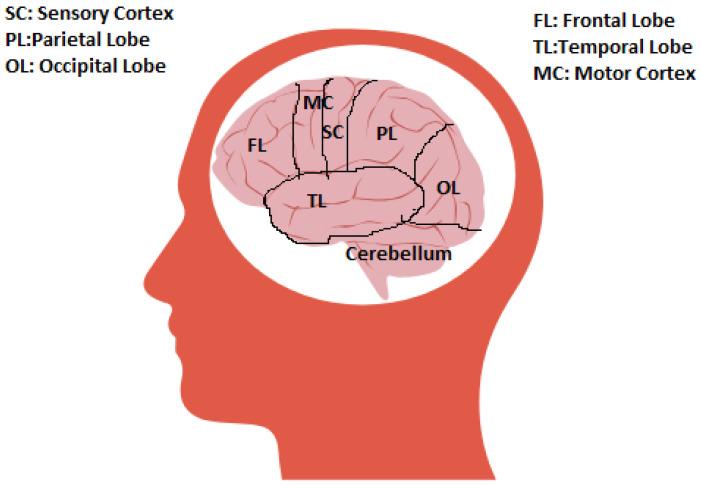
Regions of human brain.

## 4. Motor Learning

Motor imagery is intricately interconnected with motor learning, which refers to the process of acquiring and refining motor skills through repetitive practice. Current findings propose that motor imagery can augment motor learning through various mechanisms. Motor learning using motor imagery typically involves three phases Figure 3:

### 4.1. Cognitive Rehearsal

Motor imagery enables individuals to engage in mental rehearsal of movements, fostering cognitive planning and organization of actions. By repetitively engaging in the mental practice of the desired movement, individuals can optimize their motor plans, therefore enhancing accuracy, coordination, and timing [12].

### 4.2. Neural Activation

Motor imagery stimulates neural circuits analogous to those activated during physical execution, resulting in neuroplastic changes that support motor learning [16,17,18]. Repeated activation of these neural circuits through imagery can reinforce the associations and pathways related to the intended movement, ultimately contributing to skill acquisition and consolidation of motor [16].

### 4.3. Skill Transfer

Motor imagery possesses the capacity to bolster the transfer of learned skills from a cognitive domain to a physical domain. By mentally rehearsing specific movements, individuals can develop a more profound comprehension [10].

## 5. Rehabilitation Using Motor Imagery

Motor imagery has demonstrated significant potential in numerous disability conditions, providing possible advantages for functional rehabilitation. This section delves into the implementation of motor imagery in various disabilities, encompassing stroke rehabilitation, spinal cord injury, traumatic brain injury, Parkinson’s disease, cerebral palsy, and musculoskeletal disorders [18,19]. Figure 4 and Table 1 shows different use cases of MI for rehabilitation.

### 5.1. Stroke

Stroke frequently leads to motor impairments and functional restrictions. As demonstrated by a randomized, placebo-controlled study conducted by [20], motor imagery has emerged as an efficacious supplemental therapy in stroke rehabilitation. Engaging in mental practice allows stroke survivors to mentally simulate movements, encouraging neural activation and expediting motor recuperation. Researchers [21] discovered that integrating physical therapy with mental practice resulted in superior outcomes compared to conventional physical therapy alone. Moreover, studies [22,23,24] established that motor imagery can contribute to functional improvements in activities of daily living and promote cortical reorganization in stroke-affected brain regions.

### 5.2. Spinal Cord Injury

Spinal cord injuries (SCI) often engender severe motor impairments and loss of voluntary movement below the injury level. Opsommer et al. [25] observed positive effects of motor imagery training in individuals with SCI. Through visualization and mental rehearsal of movements, subjects can activate cortical and subcortical areas associated with motor control. This activation may lead to enhancements in motor function, muscle strength, and spasticity, as demonstrated by studies conducted by Opsommer et al. [25]. Furthermore, motor imagery interventions can augment body awareness, stimulate neuroplasticity, and expedite the integration of prosthetic devices or assistive technologies [26,27].

### 5.3. Traumatic Brain Injury

Traumatic brain injuries (TBI) can manifest various motor impairments, including challenges with motor coordination, muscle weakness, and balance issues. Scientific investigations have indicated that motor imagery interventions can address these obstacles and ameliorate motor outcomes in individuals with TBI. Oostra et al. [15] determined that motor imagery can bolster motor planning and coordination, facilitating the reacquisition of motor skills. This finding is corroborated by research conducted by Liu et al. [28], who proved that motor imagery training can enhance motor performance, functional mobility, and balance in subjects with TBI.

### 5.4. Parkinson’s Disease

Parkinson’s disease (PD) constitutes a neurodegenerative disorder typified by motor manifestations such as bradykinesia, tremors, and rigidity. Motor imagery has surfaced as a promising therapeutic strategy for addressing motor deficits and augmenting motor control in PD-afflicted individuals. As reported by Nicholson et al. [17], motor imagery facilitates the enhancement of motor planning, coordination, and timing for individuals with PD. This amelioration is accomplished through the mental execution of movements and activation of neural networks pertinent to motor control. Furthermore, the study revealed that motor imagery interventions contribute to improvements in gait, balance, and overall motor performance for PD patients [29,30,31].

### 5.5. Cerebral Palsy

Cerebral palsy (CP) encompasses a collection of motor disorders originating from brain damage occurring during early developmental stages. Motor imagery has garnered interest as a potential supplementary therapy for CP rehabilitation. Research conducted by Ragunath et al. [32,33] determined that individuals with CP can benefit from motor imagery interventions, which foster enhanced motor planning, coordination, and overall motor performance. By mentally rehearsing movements, individuals can stimulate neural circuits associated with motor control while expediting the integration of novel motor patterns. The study exhibited improvements in upper limb functionality, gait, and balance among CP patients [34,35].

### 5.6. Musculoskeletal Disorders

Motor imagery techniques prove advantageous for those affected by various musculoskeletal disorders, encompassing conditions such as osteoarthritis, chronic pain, and limb amputation. Investigations have substantiated the efficacy of motor imagery in palliating pain intensity, promoting joint function, and expediting rehabilitation processes in those afflicted with musculoskeletal disorders [36]. Mental rehearsal generates effective modulation of pain signals while activating cortical regions pertinent to pain processing and fostering the reorganization of pain-related neural networks.

### 5.7. Amputations

Motor imagery can facilitate the adaptation to prosthetic limbs. By mentally controlling a virtual or real prosthetic, users can reinforce neural pathways associated with limb movement [37].

**Table 1 healthcare-11-02653-t001:** Summary of Motor Imagery in Rehabilitation for Various Disabilities.

References	Disability	Key Findings in Research
[11,12,13,38,39]	Stroke	Neural activation for motor recovery Improved outcomes with mental practice Enhances activities of daily living
[14,40,41]	Spinal Cord Injury	Improves motor function and strength Enhancements in muscle strength Facilitates body awareness and neuroplasticity
[15,16]	Traumatic Brain Injury	Enhances motor planning and coordination Improved motor performance and mobility Addresses balance issues and muscle weakness
[17,42,43]	Parkinson’s Disease	Improves motor planning and coordination Enhancements in gait and balance Augments motor control and timing
[18,44,45]	Cerebral Palsy	Enhances motor planning and coordination Improved upper limb functionality Stimulates neural circuits for motor control
[19]	Musculoskeletal Disorders	Reduces pain intensity and promotes joint function Expedites rehabilitation processes Neural network reorganization

The integration of motor imagery into rehabilitation protocols produces more extensive and efficacious interventions for individuals coping with disabilities. The referenced studies corroborate the potential of motor imagery to bolster motor function, facilitate motor learning, foster neuroplasticity, and elevate overall functional outcomes in the domains of stroke rehabilitation, spinal cord injury, traumatic brain injury, Parkinson’s disease, cerebral palsy, and musculoskeletal disorders. Continued research and examination are requisite to optimize motor imagery techniques and generate customized interventions for specific disability conditions.

## 6. Effectiveness of Motor Imagery for Rehabilitation

The integration of motor imagery into rehabilitation programs can refine motor functions, amplify cognitive capacities, foster psychological well-being, and advance neuroplasticity and motor recovery. Persistent investigation of motor imagery techniques is essential to maximize their potential in rehabilitation contexts Figure 5.

### 6.1. Motor Function Improvement

#### 6.1.1. Acquisition of Motor Skills

Studies [37,38] revealed that motor imagery fosters cognitive rehearsal, subsequently leading to an augmentation of skill acquisition and motor performance.

#### 6.1.2. Motor Control and Coordination Enhancement

Hétu et al. [39] posited that motor imagery stimulates the neural networks responsible for motor planning and execution, thus improving motor control and coordination.

#### 6.1.3. Rehabilitation Targeting Specific Motor Impairments

Caligiore et al. [40] corroborated that motor imagery can address certain motor impairments, such as muscular weakness or spasticity, resulting in improved motor functionality.

**Figure 5 healthcare-11-02653-f005:**
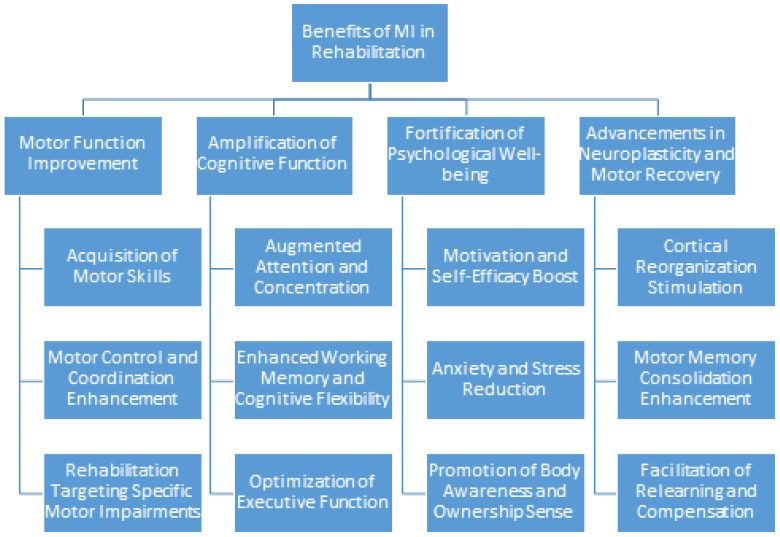
Benefits of using MI for Rehabilitation.

### 6.2. Amplification of Cognitive Function

#### 6.2.1. Augmented Attention and Concentration

Di Rienzo et al. [41] elucidated that habitual practice of motor imagery bolsters attentional control and concentration capabilities.

#### 6.2.2. Enhanced Working Memory and Cognitive Flexibility

Wright et al. [42] established that motor imagery promotes an increase in working memory capacity and cognitive adaptability.

#### 6.2.3. Optimization of Executive Function

Holmes et al. [43] emphasized the engagement of executive functions through motor imagery, leading to refinements in planning, organization, and decision-making skills.

### 6.3. Fortification of Psychological Well-Being

#### 6.3.1. Motivation and Self-Efficacy Boost

Malouin et al. [44] discovered that motor imagery intensifies motivation and self-efficacy during rehabilitation processes.

#### 6.3.2. Anxiety and Stress Reduction

Di Rienzo et al. alluded to a decrease in anxiety and stress levels among individuals partaking in rehabilitation when utilizing motor imagery [41].

#### 6.3.3. Promotion of Body Awareness and Ownership Sense

Sharma et al. [45] documented that motor imagery fosters body consciousness and a sense of personal body ownership, ultimately enhancing psychological well-being.

### 6.4. Advancements in Neuroplasticity and Motor Recovery

#### 6.4.1. Cortical Reorganization Stimulation

Guillot et al. [46] presented evidence demonstrating that motor imagery stimulates cortical regions associated with motor control, subsequently resulting in cortical reorganization and motor recovery.

#### 6.4.2. Motor Memory Consolidation Enhancement

Kundi et al. [47] substantiated that motor imagery elevates motor memory consolidation, leading to improved motor performance and recovery.

#### 6.4.3. Facilitation of Relearning and Compensation

Braun et al. [48] advocated that motor imagery aids in the relearning of motor abilities and compensation for motor impairments.

## 7. Motor Imagery Techniques and Training Protocols

Motor imagery techniques and training protocols are essential for optimizing the advantages of motor imagery in rehabilitation processes [48,49]. This section delves into various motor imagery techniques and training protocols, including mental practice, virtual reality, biofeedback and neurofeedback, electromyography (EMG), brain–computer interfaces (BCI), and the efficacy of combined approaches.

### 7.1. Mental Practice

It constitutes the basis of motor imagery, encompassing the cognitive rehearsal of movements without physical execution. It can be performed independently or facilitated by a therapist or audiovisual cues, employing diverse methods such as visualizing movements, imagining related sensations, and mentally simulating the timing and sequencing of actions [41,50]. Integrating mental practice into rehabilitation protocols as an independent technique or in combination with physical practice has demonstrated improvements in motor function, muscle strength, and coordination [34,35,48].

### 7.2. Virtual Reality

VR offers an immersive and interactive platform for motor imagery training, enabling individuals to participate in realistic and engaging simulations of movements and activities [14]. VR-based motor imagery training employs virtual environments, avatars, and interactive tasks to augment visualization and cognitive rehearsal of movements. This technology provides real-time visual and auditory feedback, enhancing presence and embodiment, and has exhibited promising outcomes in stroke rehabilitation, spinal cord injury, as well as other neurological conditions by ameliorating motor function, coordination, and quality of life [14,29].

### 7.3. Biofeedback and Neurofeedback

These techniques deliver real-time data on physiological processes associated with motor control that can be incorporated with motor imagery to boost self-awareness, refine motor performance, and foster neuroplastic changes [34,35]. Biofeedback monitors physiological signals such as heart rate, muscle activity, or skin conductance; whereas neurofeedback evaluates brain activity utilizing electroencephalography (EEG). Merging biofeedback or neurofeedback with motor imagery allows individuals to obtain instant feedback on their cognitive practice, guiding them to modify their motor imagery strategies. These techniques have been effectively implemented in stroke rehabilitation, Parkinson’s disease, and other motor disorders, resulting in enhanced motor control and functional outcomes Figure 6 [1,51,52,53].

### 7.4. Electromyography

EMG evaluates muscle activity and can be integrated with motor imagery to offer feedback on the activation of specific muscles during cognitive rehearsal. EMG-assisted motor imagery training empowers individuals to supervise and modulate muscle activity, facilitating motor learning and control. By converging EMG with motor imagery, individuals can augment muscle activation, refine coordination, and optimize motor performance [52,54,55,56].

### 7.5. Brain–Computer Interfaces

BCIs facilitate direct interactions between the cerebral cortex and external apparatuses and have been utilized for the detection and interpretation of neuronal signals correlated with motor imagery. These signals undergo a translation process to produce control commands for assistive technologies or virtual avatars, therefore enhancing neural pathways associated with motor control and fostering neuroplastic adaptations. BCIs demonstrate potential applicability in post-stroke rehabilitation, spinal cord injuries, and other medical conditions, offering novel opportunities for motor recovery and functional restoration [15,57,58].

Integrating multiple motor imagery methodologies and training regimens augments the advantages of motor imagery in rehabilitation efforts [54]. Comprehensive paradigms, such as the amalgamation of mental practice with virtual reality, biofeedback, and electromyography, yield extensive and interactive training programs that address diverse components of motor functionality and neuronal activation. These synergistic approaches afford a customizable and adjustable rehabilitation experience, fostering multisensory integration, engagement, and motivation—all crucial elements in motor learning and recuperation.

Motor imagery techniques and training protocols encompassing mental practice, VR, biofeedback, neurofeedback, EMG, and BCI offer distinct benefits in reinforcing motor function, accelerating neuroplasticity, and ameliorating overall rehabilitation outcomes. Incorporating these methodologies into individualized and tailored rehabilitation initiatives can enhance the efficacy of motor imagery interventions for disabled individuals. Further exploration and technological advancements are imperative to unleashing the full potential of these methods and expediting their widespread integration within clinical environments.

## 8. Challenges of MI Based Rehabilitation

The potential of motor imagery in rehabilitation is significant; however, numerous challenges and limitations must be acknowledged. This section examines the obstacles relating to variability in imagery ability, individual disparities and training responses, incorporation of motor imagery into clinical practice, and technological restrictions.

### 8.1. Variability in Imagery Proficiency

A primary challenge when employing motor imagery is the varying proficiency in individuals’ ability to imagine movements. Some individuals possess a heightened capability to create vivid and detailed mental images of actions, while others encounter difficulties generating distinct mental representations. This variability may hinder the effectiveness of motor imagery interventions, as those with limited imagery proficiency may face challenges engaging in precise and immersive mental simulations.

To tackle this issue, customized strategies are necessary. Practitioners and researchers must assess and comprehend each person’s imagery proficiency to design personalized motor imagery training programs. Techniques such as offering visual prompts, auditory guidance, or employing guided imagery can assist individuals with diminished imagery capabilities.

### 8.2. Individual Disparities and Training Response

Cognitive and neural process heterogeneity can impact an individual’s reaction to motor imagery training. Elements like attentional capacity, cognitive adaptability, and baseline motor functioning can influence the efficacy of motor imagery interventions. Furthermore, psychological aspects, motivation levels, and emotional states may also sway an individual’s engagement during motor imagery exercises.

Recognizing and accommodating these individual disparities is crucial for maximizing motor imagery intervention outcomes. Tailored training initiatives that take into account cognitive, emotional, and motivational traits can boost engagement and elicit improved responses to training. Additionally, continuous evaluation and modification of training protocols based on individual progress can guarantee the effectiveness of motor imagery interventions.

### 8.3. Motor Imagery Implementation in Clinical Practice

Incorporating motor imagery into clinical practice presents challenges due to multiple factors. One predominant obstacle involves insufficient awareness and knowledge regarding motor imagery among healthcare professionals. Educating healthcare providers on motor imagery principles and applications is vital for its successful integration into rehabilitation programs.

Another hurdle is the constraints related to time and resources in clinical environments. Integrating motor imagery into existing rehabilitation protocols may necessitate supplementary time and specialized training for therapists. Moreover, the accessibility of suitable technology and equipment for motor imagery training can pose limitations in certain clinical settings. To surmount the aforementioned obstacles, it is imperative to introduce educational programs aimed at increasing awareness and offering specialized training on motor imagery for healthcare practitioners. The creation of cost-effective and accessible technologies tailored for motor imagery instruction will further streamline its incorporation into clinical procedures.

### 8.4. Technological Constraints

The implementation of motor imagery interventions can be hindered by various technological constraints. For instance, the acquisition of virtual reality systems demands substantial investments, specialized knowledge, and physical space. Therefore, it is essential to develop economical and intuitive virtual reality platforms to render them more readily available in clinical environments.

Moreover, the precision and dependability of measurements and feedback during motor imagery training may be restricted due to technical factors. Electroencephalography (EEG) signals are prone to noise interference and artifacts, which compromise the quality of neurofeedback-centric training. Likewise, electromyography readings may exhibit limitations in discerning minor alterations in muscular activity occurring during motor imagery.

Progression in technology, including enhancements in EEG and EMG apparatuses, signal processing methodologies, and real-time feedback systems, is vital to combat these constraints. Sturdy and dependable technologies delivering precise feedback and objective evaluations of motor imagery performance have the potential to augment the efficacy and overall efficiency of motor imagery interventions.

Despite the considerable promise of motor imagery in the realm of rehabilitation, several challenges remain to be confronted. The fluctuation in imagery aptitude, individual distinctions, integration into clinical practice, and technological barriers contribute to hindrances against broader adoption. By tackling these issues through customized approaches, pedagogical initiatives, and technological advancements, motor imagery can be effectively employed to enhance rehabilitation outcomes for persons with disabilities. Persistent research efforts and interdisciplinary collaboration between scholars, clinical professionals, and technologists are indispensable for overcoming these obstacles and fully exploiting the capabilities of motor imagery within the sphere of rehabilitation.

## 9. Future Directions

Motor imagery in rehabilitation is an ever-evolving domain, with multiple promising avenues for future exploration and enhancement. This section delves into the crucial areas poised to advance motor imagery interventions, including the creation of individualized training curricula, refining neurofeedback methodologies, breakthroughs in virtual reality and brain–computer interface (BCI) technology, long-standing follow-ups and maintenance strategies, as well as cross-disciplinary collaborations.

### 9.1. Development of Personalized Training Programs

Developing customized training programs catering to specific individual requirements and traits is imperative for augmenting the efficacy of motor imagery interventions. Upcoming research must emphasize the formulation of assessment instruments and models that allow precise evaluation of an individual’s cognitive, emotional, and motor aspects to steer the development of specialized motor imagery protocols. It is vital to consider components such as imagery competence, attention capacity, foundational motor functionality, and motivational influences to certify maximum involvement and effective training outcomes.

Moreover, advancements in machine learning and artificial intelligence may expedite the establishment of intelligent systems that adapt based on individual progress and instantaneous feedback. These systems can dynamically modify difficulty levels, material content, and evaluation modalities to optimize training experiences and facilitate improved results.

### 9.2. Optimization of Neurofeedback Techniques

Therein lies immense potential in employing neurofeedback methodologies to boost the efficacy of motor imagery interventions. Future research directions should concentrate on enhancing neurofeedback paradigms by ensuring accurate and dependable real-time brain activity feedback. Improvements in signal processing algorithms, feature extraction techniques, and machine learning methodologies will facilitate more robust detection and interpretation of neural markers associated with motor imagery.

Furthermore, investigating innovative neurofeedback modalities beyond conventional EEG applications (e.g., functional near-infrared spectroscopy (fNIRS) or magnetoencephalography (MEG)) can furnish supplementary data and increase neurofeedback training specificity. The incorporation of numerous modalities coupled with advanced data analytics can significantly improve the precision and efficacy of neurofeedback-centered motor imagery interventions.

### 9.3. Advancements in Virtual Reality and BCI Technologies

With rapid progression in the fields of virtual reality and brain–computer interface technology, immense potential exists for motor imagery. Future research should concentrate on aspects such as the enhancement of virtual reality platform fidelity, usability, and accessibility for motor imagery training purposes. Essential elements include improvement in virtual environmental realism, development of user-centric interfaces, and minimizing expenses and technical intricacies associated with VR systems. Consequently, progressive developments in BCI technology have the potential to profoundly impact the incorporation of motor imagery into rehabilitative practices. Enhancements in the precision and dependability of BCI systems, investigation of non-invasive and wearable alternatives, and refinement of neural signal translation into motor instructions can pave the way for novel opportunities in motor recuperation and assistive technologies. Merging BCI advancements with virtual reality platforms can generate immersive, interactive training environments that bolster user engagement and encourage neuroplasticity.

### 9.4. Prolonged Monitoring and Sustained Progress

Extensive monitoring and sustained progress in motor imagery interventions are vital for evaluating long-lasting effects and ensuring ongoing improvement. Future studies ought to scrutinize the enduring advantages and maintenance of motor function enhancements attained via motor imagery exercises. This encompasses determining the ideal dosage, scheduling, and duration of training sessions to achieve durable outcomes.

Additionally, advocating for self-management and home-based motor imagery programs may augment the durability and availability of these interventions. Mobile applications, wearable technology, and telerehabilitation platforms can support remote supervision, direction, and feedback—allowing individuals to maintain their motor imagery practice beyond clinical environments.

### 9.5. Synergy and Comprehensive Methodologies

Synergistic endeavors and all-encompassing methodologies are essential to advance the realm of motor imagery within rehabilitation contexts. Researchers, healthcare professionals, technologists, and individuals with disabilities should collaborate to stimulate innovation, share expertise, and bridge divides between research findings and clinical applications. Collective efforts can result in the creation of comprehensive, versatile motor imagery interventions tailored to address the wide-ranging needs and obstacles confronted by those living with disabilities.

Furthermore, comprehensive collaborations can facilitate the integration of motor imagery with additional rehabilitative strategies such as physical therapy, occupational therapy, and cognitive training sessions. Combined interventions designed to synergistically target motor skills alongside cognitive capacity and psychological factors could potentially augment overall rehabilitative outcomes—fostering an enhanced quality of life for individuals with disabilities. Persistent research, innovation, and collaboration are indispensable in propelling the field forward and ameliorating the lives of individuals utilizing motor imagery in rehabilitation.

## 10. Conclusions

In this review, we delved into the concepts, mechanisms, applications, benefits, techniques, challenges, and prospective directions of motor imagery within the framework of disability rehabilitation. Motor imagery encompasses the mental rehearsal of movements without physical execution and stimulates comparable brain regions to those activated during physical movement. This process can augment motor learning and performance via neuroplasticity mechanisms. Motor imagery boasts a broad spectrum of applications across various disabilities—from stroke rehabilitation and spinal cord injury to traumatic brain injury, Parkinson’s disease, cerebral palsy, and musculoskeletal disorders. The technique harbors the potential for enhancing motor functionality, coordination, and overall quality of life for affected populations.

Motor imagery offers several advantages in rehabilitation efforts by improving motor function, cognitive capabilities, and psychological well-being while fostering neuroplasticity and motor recovery in disabled individuals [59,60,61]. A myriad of techniques and training protocols can be employed to support motor imagery interventions—including mental practice, virtual reality applications, biofeedback- and neurofeedback-based technologies, electromyography, brain–computer interfaces (BCI), and integrated methods customizable to individual needs. Each approach presents distinctive benefits that can amplify motor function while optimizing rehabilitative outcomes. Nonetheless, several challenges accompany motor imagery interventions—such as variability in imagery ability, individual differences and responsiveness to the training protocols, incorporation into standard clinical practice, and technological limitations. Addressing these challenges through tailored methodologies, targeted education, and advancements in relevant technologies is paramount in overcoming constraints and promoting comprehensive implementation. The prospects of motor imagery in rehabilitation are contingent on the evolution of personalized training paradigms, fine-tuning of neurofeedback techniques, progress in virtual reality- and BCI-based systems, sustained follow-up measures, alongside teamwork and multidisciplinary cooperation. These domains exhibit potential for augmenting the efficacy and accessibility of motor imagery interventions. The article delivered an exhaustive analysis of motor imagery in disability rehabilitation, emphasizing its potential influence on enhancing motor function, cognitive function, psychological well-being, and overall quality of life for individuals with disabilities. The findings propose that motor imagery interventions can be incorporated into clinical practice to supplement traditional rehabilitation methodologies and improve treatment outcomes. The prospects for motor imagery in rehabilitation appear promising. The formulation of personalized training schemes, refinement of neurofeedback methods, progress in virtual reality and BCI technologies, and long-term follow-up may contribute to more efficient and sustainable interventions. Collaborative efforts and multidisciplinary approaches are crucial for fostering innovation, connecting research and clinical practice, and tackling the challenges encountered in implementing motor imagery interventions. The potential ramifications of motor imagery surpass rehabilitation settings; it can enhance the lives of individuals with disabilities by equipping them with self-management techniques, boosting their autonomy, and promoting community integration. In summation, motor imagery serves as a valuable instrument in disability rehabilitation with an extensive array of applications and advantages. Through further exploration of its potential, addressing impediments, and capitalizing on technological advancements and interdisciplinary collaborations, the full potential of motor imagery interventions can be harnessed to revolutionize the realm of rehabilitation for disabled individuals.

## Figures and Tables

**Figure 1 healthcare-11-02653-f001:**
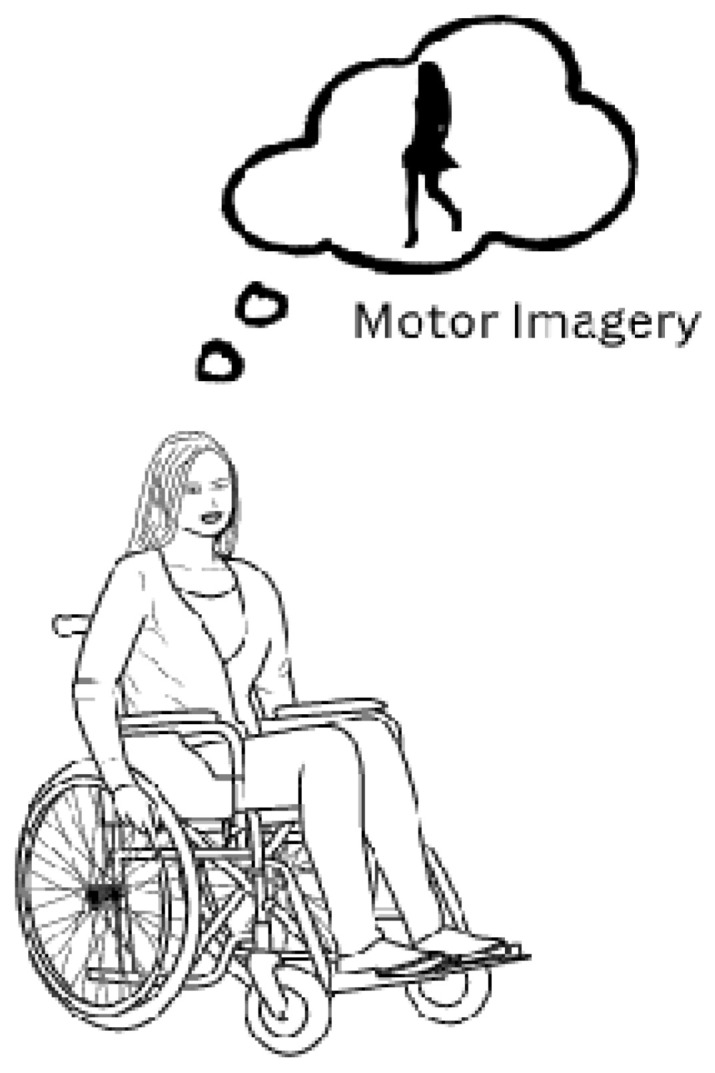
Motor imagery.

**Figure 3 healthcare-11-02653-f003:**
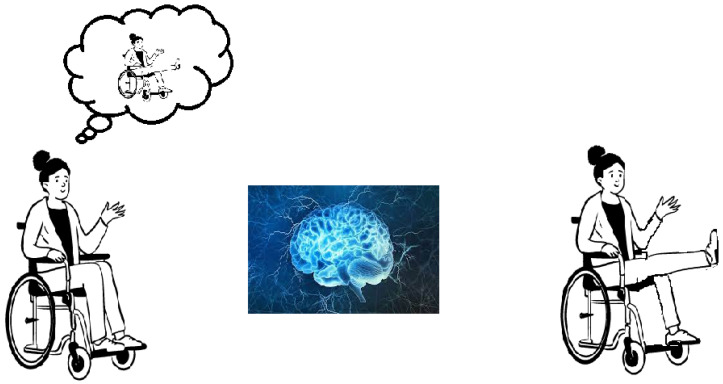
Motor Learning Phases: Cognitive rehearsal, Neural activation and Skill Transfer.

**Figure 4 healthcare-11-02653-f004:**
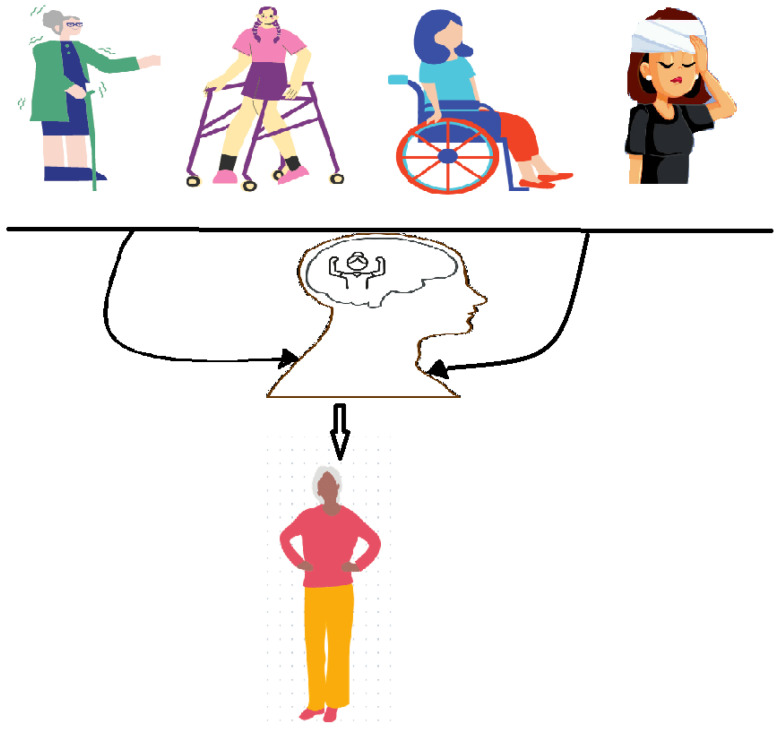
Rehabilitation using motor imagery.

**Figure 6 healthcare-11-02653-f006:**
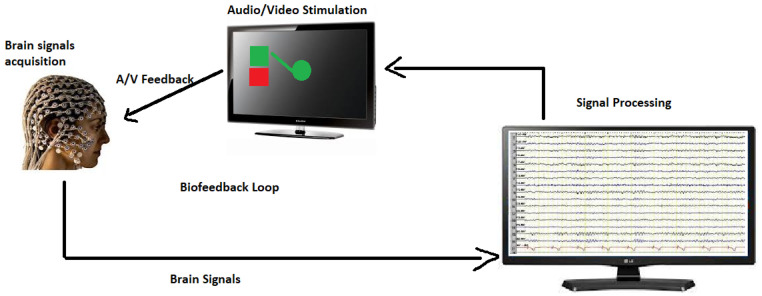
Biofeedback loop Therapy.

## Data Availability

Not applicable.

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
