# Peer review of "Leveraging Motor Imagery Rehabilitation for Individuals with Disabilities: A Review"

_healthcare, 2023, doi:10.3390/healthcare11192653_

Round 1
Reviewer 1 Report
Thank you for the opportunity to review this manuscript. This was extremely interesting. I have only a few minor recommendations. Page 1 line 19 and page 3 line 61, may consider changing "comprehension" to understanding.
Also, would like to see search method included, ie databases, keywords, etc so that this can be replicated.
Author Response
|
1 |
Page 1 line 19 and page 3 line 61, may consider changing "comprehension" to understanding. |
Both identified lines have be rephrased to make these more readable. |
|
|
would like to see search method included, i.e. databases, keywords, etc |
Used keywords such as motor imagery, reviews, rehabilitation etc. |
Reviewer 2 Report
After reading the manuscript, verify some errors such as:
• Lack of parentheses in the numbers of the bibliography introduced in the text, as in the lines: 18, 22, 28…..
• The introduction is structured, this cannot be the objectives at the beginning… they are always at the end
• I can't find the figures stated in the text….
• I am missing a flowchart of the selected items
• There are no tables explaining what each author of the subject says….
• You need a big return to be accepted
Author Response
|
Lack of parentheses in the numbers of the bibliography introduced in the text, as in the lines: 18, 22, 28….. |
Parentheses have been added for references |
|
The introduction is structured, this cannot be the objectives at the beginning… they are always at the end. |
Required changes in structure of introduction has been made |
|
I can't find the figures stated in the text…. |
All figures’ references has been added |
|
There are no tables explaining what each author of the subject says…. |
Table has been added |
Reviewer 3 Report
It is an interesting and timely review.
Some comments and instructions:
1) Figure 1 is very pixelated. The image must be improved. Also the balloon in Figure 3 and the central image in Figure 4 are pixelated; must be improved. There is a dot in the bottom image of Figure 4; It would be convenient to eliminate it.
2) You have to leave double spacing under the figures' feet.
3) It would be very convenient to add to the drawing in Figure 2 the acronyms corresponding to the brain areas specified in the paragraph: Regions such as the primary motor cortex (M1) and premotor cortex, responsible for motor planning and execution, are activated during motor imagery sequences. These regions generate neural signals analogous to those produced during actual movements. The supplementary motor area (SMA) also participates in this process, contributing to the initiation and coordination of motor plans. Parietal regions like the superior parietal lobule and inferior parietal lobule play a crucial role in sensorimotor integration during motor imagery. These regions combine visual, proprioceptive, and kinesthetic information to formulate a coherent mental representation of the movement [9]. The Mirror Neuron System (MNS)
For example, add something similar to figure 1 of the following article: Acharya, S., & Shukla, S. (2012). Mirror neurons: Enigma of the metaphysical modular brain. Journal of natural science, biology, and medicine, 3(2), 118–124. https://doi.org/10.4103/0976-9668.101878
4) Vancouver Style Standards, AMA and MDPI require consecutive numbering of bibliographic references.
Therefore, the following paragraphs must correct their numbering (and please review the bibliographic numbering of the entire manuscript):
- Furthermore, studies [13,38,39] established that motor… (line 146)
- expedite integration of prosthetic devices or assistive technologies [40,41] . (line 157)
- motor performance for PD patients[42,43]. (line 176)
- (lines 186, 206, 214, etc.)
5) Figure 5, is it your own creation? Or is it taken from a publication? (in this case, it should be referenced in the bottom of the figure)
6) I find it interesting that they also made some comments about the practice of motor images without technology, as it was in the beginning, or as it sometimes still is in low-tech environments or with few resources.
7) Most of the first 31 bibliographical references are more than 5 years old. An increase in the percentage of more recent references would be appreciated.
For example, add: Silva S, Borges LRDM, Santiago L, Lucena L, Lindquist AR, Ribeiro T. Motor imagery for gait rehabilitation after stroke. Cochrane Database of Systematic Reviews 2020, Issue 9. Art. No.: CD013019. DOI: 10.1002/14651858.CD013019.pub2.
https://www.cochranelibrary.com/cdsr/doi/10.1002/14651858.CD013019.pub2/full
And review (in case you can add or replace any significant reference): https://pubmed.ncbi.nlm.nih.gov/?term=Motor+Imagery+Rehabilitation&filter=datesearch.y_5&sort=date
8) It would also be convenient to add some comment to the work with motor images in childhood.
Saleem G.T. (2023). Defining and measuring motor imagery in children: mini review. Frontiers in psychology, 14, 1227215. https://doi.org/10.3389/fpsyg.2023.1227215
Author Response
|
Figure 1 is very pixelated. The image must be improved. Also the balloon in Figure 3 and the central image in Figure 4 are pixelated; must be improved. There is a dot in the bottom image of Figure 4; It would be convenient to eliminate it. |
All Figures have been improved as per comments. |
|
You have to leave double spacing under the figures' feet. |
Changes have been made. |
|
It would be very convenient to add to the drawing in Figure 2 the acronyms corresponding to the brain areas specified in the paragraph: Regions such as the primary motor cortex (M1) and premotor cortex, responsible for motor planning and execution, are activated during motor imagery sequences. These regions generate neural signals analogous to those produced during actual movements. The supplementary motor area (SMA) also participates in this process, contributing to the initiation and coordination of motor plans. Parietal regions like the superior parietal lobule and inferior parietal lobule play a crucial role in sensorimotor integration during motor imagery. These regions combine visual, proprioceptive, and kinesthetic information to formulate a coherent mental representation of the movement [9]. The Mirror Neuron System (MNS). For example, add something similar to figure 1 of the following article: Acharya, S., & Shukla, S. (2012). Mirror neurons: Enigma of the metaphysical modular brain. Journal of natural science, biology, and medicine, 3(2), 118–124. https://doi.org/10.4103/0976-9668.101878 |
Figure 2 has been re drawn. |
|
Vancouver Style Standards, AMA and MDPI require consecutive numbering of bibliographic references. Therefore, the following paragraphs must correct their numbering (and please review the bibliographic numbering of the entire manuscript):
- Furthermore, studies [13,38,39] established that motor… (line 146)
- expedite integration of prosthetic devices or assistive technologies [40,41] . (line 157)
- motor performance for PD patients[42,43]. (line 176)
- (lines 186, 206, 214, etc.) |
Required changes have been made. Numbering is now in proper order as per first citation of the reference. |
|
Figure 5, is it your own creation? Or is it taken from a publication? (in this case, it should be referenced in the bottom of the figure) |
Figure 5 is created by me. It is not taken from any other article |
|
I find it interesting that they also made some comments about the practice of motor images without technology, as it was in the beginning, or as it sometimes still is in low-tech environments or with few resources. |
|
|
Most of the first 31 bibliographical references are more than 5 years old. An increase in the percentage of more recent references would be appreciated.
For example, add: Silva S, Borges LRDM, Santiago L, Lucena L, Lindquist AR, Ribeiro T. Motor imagery for gait rehabilitation after stroke. Cochrane Database of Systematic Reviews 2020, Issue 9. Art. No.: CD013019. DOI: 10.1002/14651858.CD013019.pub2.
https://www.cochranelibrary.com/cdsr/doi/10.1002/14651858.CD013019.pub2/full
And review (in case you can add or replace any significant reference): https://pubmed.ncbi.nlm.nih.gov/?term=Motor+Imagery+Rehabilitation&filter=datesearch.y_5&sort=date
8) It would also be convenient to add some comment to the work with motor images in childhood.
Saleem G.T. (2023). Defining and measuring motor imagery in children: mini review. Frontiers in psychology, 14, 1227215. https://doi.org/10.3389/fpsyg.2023.1227215 |
Bibliography is having a mix of references to classical to state of the art articles as for a comprehensive review this blend is significant. However more references have also been added. |
Reviewer 4 Report
Thank you for your manuscript. It is interesting but you have some important questions to answer. Please see my comments below…
P1L15. “Conventional rehabilitation methods such as physiotherapy and assistive apparatuses have demonstrated efficacy, but often encounter limitations in achieving optimal results.” – Reference…
2nd paragraph – This paragraph makes no sense in this part of the introduction section. The aim should be presented at the end of the introduction section...
P1L22. “...process for individuals with disabilities.” – It is important to define what you define as individuals with disabilities…
P2 – I don't see any relevance of figure 1…
P2L32. “Neuroimaging studies…” – Which are these studies?
P2L41. “Through mental rehearsal of movements, individuals can enhance their motor planning, coordination, and precision, resulting in more proficient motor execution [9].” – I believe that is important to refer that this reference is a systematic review…
P3L47. “The advantages of motor imagery in enhancing motor functionality, cognitive capability augmentation, psychological well-being promotion, and fostering neuroplasticity for motor recuperation will be examined [7]. Additionally, the manuscript will delve into diverse motor imagery techniques and training protocols including mental practice, virtual reality applications, biofeedback modalities, and brain-computer interfaces while analyzing their respective strengths and drawbacks [9].” – It is not clear why references were included in these sentences…
P3 – The last three paragraphs should form just one paragraph. And it is convenient in this last paragraph to summarize what will be the next points of the article…
P3L67. “This allows individuals to generate and manipulate mental representations of motor actions, incorporating sensory, perceptual, and proprioceptive elements.” – Reference…
P4L94. “Neuroimaging studies…” – Which are these studies?
Where are the images referenced in the text?
P4L98. “Regions such as the primary motor cortex (M1) and premotor cortex, accountable for motor planning and execution, are activated during motor imagery sequences.” – References…
Motor Learning section – According to Figure 3, the points defined in the Motor Lerning section correspond to phases... But this is not perceptible in the text…
P4L124. “Motor imagery stimulates neural circuits analogous to those activated during physical execution, resulting in neuroplastic changes that support motor learning.” – References…
For me, the need to divide points 6 and 7 into two different points is not clear... Why not just be one point in the article?
From points 6 and 7, is it possible to conclude that more studies are needed? The answer is not clear in the text...
P9L256.“…virtual reality (VR)…” – On page 3, "virtual reality" had already appeared...
P10L305.“…virtual reality (VR)…” – and appear also in pages 11, 12, and 13…
P10L306.“… electromyography (EMG)…” – had already been referenced on the previous page…
P11L371.“… electromyography (EMG)…” – referenced again…
Points 9 and 10 are just based on the authors' opinion... I believe that for a comprehensive review it doesn't make sense...
No comments...
Author Response
|
3 |
P1L15 “Conventional rehabilitation methods such as physiotherapy and assistive apparatuses have demonstrated efficacy, but often encounter limitations in achieving optimal results.” – Reference… |
References added |
|
|
2nd paragraph – This paragraph makes no sense in this part of the introduction section. The aim should be presented at the end of the introduction section... |
Suggested change has been made |
|
|
P1L22. “...process for individuals with disabilities.” – It is important to define what you define as individuals with disabilities… |
Definition has been added in the text. |
|
|
P2 – I don't see any relevance of figure 1… |
Figure 1 is just providing general idea of motor imagery. |
|
|
P2L32. “Neuroimaging studies…” – Which are these studies?
|
References are added. |
|
|
P2L41. “Through mental rehearsal of movements, individuals can enhance their motor planning, coordination, and precision, resulting in more proficient motor execution [9].” – I believe that is important to refer that this reference is a systematic review…
|
This has been included in the text. |
|
|
P3L47. “The advantages of motor imagery in enhancing motor functionality, cognitive capability augmentation, psychological well-being promotion, and fostering neuroplasticity for motor recuperation will be examined [7]. Additionally, the manuscript will delve into diverse motor imagery techniques and training protocols including mental practice, virtual reality applications, biofeedback modalities, and brain-computer interfaces while analyzing their respective strengths and drawbacks [9].” – It is not clear why references were included in these sentences…
|
References has been removed |
|
|
P3 – The last three paragraphs should form just one paragraph. And it is convenient in this last paragraph to summarize what will be the next points of the article…
|
Suggested changes have been made. |
|
|
P3L67. “This allows individuals to generate and manipulate mental representations of motor actions, incorporating sensory, perceptual, and proprioceptive elements.” – Reference…
|
References have been added |
|
|
P4L94. “Neuroimaging studies…” – Which are these studies?
|
References have been added |
|
|
Where are the images referenced in the text?
|
All Figures are now referenced in text |
|
|
P4L98. “Regions such as the primary motor cortex (M1) and premotor cortex, accountable for motor planning and execution, are activated during motor imagery sequences.” – References…
|
References have been added. |
|
|
Motor Learning section – According to Figure 3, the points defined in the Motor Lerning section correspond to phases... But this is not perceptible in the text…
|
Changes have been made to bring Clarity in related text. |
|
|
P4L124. “Motor imagery stimulates neural circuits analogous to those activated during physical execution, resulting in neuroplastic changes that support motor learning.” – References…
|
References have been added. |
|
|
For me, the need to divide points 6 and 7 into two different points is not clear... Why not just be one point in the article?
|
Point 6 is to describe use cases of motor imagery based rehabilitation whereas section 7 is for elaborating that how motor imagery based rehabilitation is effective. |
|
|
From points 6 and 7, is it possible to conclude that more studies are needed? The answer is not clear in the text...
|
I think like any other latest trends in technology, motor imagery has the potential to get explored more use cases. |
|
|
P9L256.“…virtual reality (VR)…” – On page 3, "virtual reality" had already appeared...
|
Changes have been made in text. |
|
|
P10L305.“…virtual reality (VR)…” – and appear also in pages 11, 12, and 13…
|
Changes have been made in text. |
|
|
P10L306.“… electromyography (EMG)…” – had already been referenced on the previous page…
|
Changes have been made in text. |
|
|
P11L371.“… electromyography (EMG)…” – referenced again…
|
Changes have been made in text. |
|
|
Points 9 and 10 are just based on the authors' opinion... I believe that for a comprehensive review it doesn't make sense... |
These sections are added to conclude and summarize the key takeaways of the article. |
Round 2
Reviewer 2 Report
Ok
Ok
Reviewer 4 Report
No comments...
No comments...